

# Electromagnetic field observations by the DEMETER satellite in connection with the 2009 L'Aquila earthquake

Bertello Igor[1], Mirko Piersanti[2,3], Maurizio Candidi[1], Piero Diego[1], Pietro Ubertini[1]

[1]INAF – Istituto di Astrofisica e Planetologia Spaziali, Roma, Italy
[2]INFN – University of Rome Tor-Vergata, Rome, Italy
[3]Consorzio Area di Ricerca in Astrofisica. L'Aquila, Italy

*Correspondence to*: mirko.piersanti@aquila.infn.it

**Abstract.** To define a background in the electromagnetic emissions above seismic regions, it is necessary to define the statistical distribution of the wave energy in absence of seismic activity and any other anomalous input (e.g. solar forcing). This paper presents a completely new method to determine both the environmental and instrumental background applied to the entire DEMETER satellite electric and magnetic field data over L'Aquila. Our technique is based on a new data analysis tool called ALIF (Adaptive Local Iterative Filtering, [Cicone et al. 2016 and 2017; Piersanti et al. 2017]). To evaluate the instrumental noise, we performed a multiscale statistical analysis, in which the instantaneous relative energy ($\epsilon_{rel}$), kurtosis and Shannon entropy was calculated. To estimate the environmental noise, a background map, divided into 1° x 1° latitude/longitude cells, of the averaged relative energy ($\overline{\epsilon_{rel}}$) has been constructed, taking into account the geomagnetic activity conditions, the presence of seismic activity and the local time sector of the satellite orbit. Any distinct signal different (over a certain threshold) from both the instrumental and environmental backgrounds will be considered as a case event to be investigated. Interestingly, on April 4, 2009, when DEMETER flew exactly over L'Aquila at UT=20:29, an anomalous signal was observed at 333 Hz on both the electric and magnetic field data, whose characteristics seem to be related to pre-seismic activity.

## 1 Introduction

The nature of the Earth's interior, in terms of the dynamic of the crust, mantle and core, can be investigated through extended ground-based and space observations [Bell et al., 1982]. In the same way, it is possible to inquest the natural or anthropogenic origin of the electromagnetic emissions (EME) using measurements in the near Earth space [Parrot and Zaslavski, 1996, Buzzi, 2006]. In addition, several works [Parrot, 1995; Parrot and Zaslavski, 1996; Buzzi, 2006] remarked that both EMEs of anthropogenic (such as, HF broadcasting stations, VLF transmitters, and so on), and of natural (i.e. Earth surface) origin can influence the dynamics and the composition of the ionosphere-magnetosphere region [Parrot, 1995; Parrot and Zaslavski, 1996]. Taking into account that extreme reliability is needed to call for preseismic phenomena, a characteristic background for the regions on Earth where we want to detect the effects of earthquake related EME should be available. *In the interim*, the comprehensive study of both magnetospheric and ionospheric disturbances driven by ground preseismic EME waves have to be carried on. In this context, aseismic fault creep and EME waves are expected to be the



principal mechanical and electromagnetic earthquake precursors, respectively [Buzzi, 2006 and reference therein]. Parrot [1995] and Parrot and Zaslavski [1996] achieved the first promising result analysing rare EME waves observations over the ionosphere-magnetosphere region. In their studies, they first discriminated between the internal and external component of the geomagnetic field and, then, they gave an efficient measure of the electric field, the plasma temperature and density of

the ionosphere. Despite several theoretical models have been developed, the physical mechanisms leading to the observation of these effect both at ground and in space are yet largely unexplained. Namely, it remains to be understood: the genesis of EME wave over the focal area (especially soon before a seismic event, if any); its propagation through lithospheric layers characterized by fixed vertical conductivity; its access into both neutral atmosphere and ionosphere; its arrival into the magnetosphere and its relative interaction. It is worth noticing that, when a EME wave propagates through both the

ionosphere and magnetosphere, the medium have to be considered dispersive, too [Chen, 1977].

The search for ionospheric disturbances associated with earthquakes relies on thorough statistical studies to disentangle seismic effects from the variations induced by the physical processes that control the ionosphere dynamic and natural emissions. Several studies have been performed, in the frame of the DEMETER (Detection of Electro-Magnetic Emissions

Transmitted from Earthquake Regions) mission, and one of them has shown a decrease of the ELF wave intensity in the frequency range between 1 and 2 kHz a few hours before the shock [Parrot et al., 2006; Píša et al., 2012 and 2013; Zhang et al., 2012; Walker et al., 2013]. Němec et al. [2008], built a statistical map of electromagnetic wave intensity obtained from DEMETER satellite ICE and IMSC data available at that time (2004-2007). Then, they estimated the probability of occurrence during a seismic event of signals with higher intensity with respect to the background level defined by the map.

Their study was the first attempt to generate a background map in the electromagnetic emission above seismic regions for the determination of the statistical distribution of the wave energy in absence of seismic activity.
The present paper uses a completely new method to determine both the environmental and instrumental background in the electromagnetic emission above seismic regions by using the DEMETER satellite electric and magnetic field observations. This algorithm is based on a new data analysis technique called ALIF (Adaptive Local Iterative Filtering, [Cicone et al. 2016

and 2017; Piersanti et al. 2018]) and through a multiscale statistical analysis of the electromagnetic observations. The results obtained with this technique allowed the construction of an electromagnetic energy background map over L'Aquila seismic region from 2004 to 2011. In addition, on April 4, 2009, two days before the 6.3 Mw earthquake [USGS Earthquake catalogue], when DEMETER flew exactly over L'Aquila at UT=20:29, an anomalous signal with respect to the background was observed.




## 2. Data and Methods

### 2.1 Demeter data

In our study, we used the data from the French satellite DEMETER, launched in 2004 on a sun-synchronous orbit at about
700km of altitude. The orbits of the satellite had an inclination of 98° and a local time of 10.30 on the day-side and 22.30 on
the night-side. The instruments were operational at geomagnetic latitudes between -65° to +65° thus providing a good
coverage of the Earth's seismic zones [Parrot et al., 2005]. The data from the electric field experiment ICE [Berthelier et al.,
2005] and the magnetic field experiment IMSC [Parrot et al., 2005] were used in order to detect any electromagnetic waves.
Among the 4 available DEMETER channels we selected the ELF band, the only one which provides the waveform of the
three components of the fields. The ELF specification is a data range in frequency from 15 Hz to 1 kHz, with a sample rate
of 2.5kHz. Due to the high data transfer resources required, the ELF acquisition is operational only in 'burst mode', so data
are available only in a fraction of the entire orbit (see Figure 1). The DEMETER mission lasted from 2004 to March 2011, so
we have a large dataset of almost 7 years of the satellite observations, altogether representing 71730 half-orbits (35865 in the
day-side and 35865 in the night-side). Within the entire available dataset, we selected the orbits with ELF data that covered
this area.

### 2.2 Adaptive local Iterative Filtering (ALIF)

The algorithm for the evaluation of both environmental and instrumental background in the electromagnetic emission above
seismic regions is based on a recent data analysis technique called ALIF, developed by Cicone et al. [2016] and Piersanti et
al. [2018]. ALIF is able, through a time-frequency analysis, to identify and quantify the variations across different scales for
non—stationary signals due to the complexity and non-linearity of the system that generated them. The reason for using this
technique is that, unlike typical data analysis methods, such as Fast Fourier transform, Wavelet and so on, ALIF does not
suffer for both limited resolution [Cohen, 2001], and interferences in the time-frequency domain [Flandrin, 1998]. Thus,
ALIF does not require any further processing of the representation. The key idea behind this method, very similar to the
Empirical mode decomposition (EMD, Huang et al. [1998]), is a "divide et impera" approach. In fact, ALIF first
decomposes a signal into several functions oscillating around zero and characterized by frequencies variable with time
(Intrinsic Mode Functions - IMFs). Then, for each IMF, it performs a time-analysis. The great difference with EMD is that
ALIF has a strong mathematical structure which guarantees the convergence and stability of the algorithm, which in turn
guarantee the physical significance of the decomposition [Piersanti et al., 2018].

### 2.3 Multiscale statistical analysis and SM test.

In order to evaluate the instrumental and environmental background of a signal s(t) (such as the magnetic and electric field
observations), we study its multiscale proprieties. To accomplish this task, we first use ALIF to decompose s(t) into
functions $IMF_\ell(t)$, characterized by a peculiar scale of variability $\ell$ [Wernik, 1997], so that:



$$s(t) = \sum_{\ell=1}^{m} IMF_\ell(t) + r(t)$$

where r(t) is the residue of the decomposition. The connection between each IMF and the scale of variability $\ell$ of s(t) has been analyzed by using the Flandrin et al. [1998] technique: A dataset characterized by an evident scale separation can be decomposed into two contributions:

$$s(t) = s_0(t) + \delta s(t)$$

where $s_0(t)$ is named baseline and $\delta s(t)$ represents the variations around the baseline. To identify $\delta s(t)$, we applied the method
proposed by Alberti et al. [2016], by defining $\delta s(t)$ as the reconstruction of a subset $s_1$ of k < m modes,

$$\delta s(t) = \sum_{\ell=1}^{k} IMF_\ell(t) \qquad \text{Eq. 1}$$

characterized by a standardized mean (i.e. the mean divided by the standard deviation) SM ≈ 0 and by IMF fluctuating at higher frequency.

Figure 2 shows an example of the application of the SMT to the DEMETER magnetic field observations (upper panel) over
L'Aquila (λ=42.334°, φ=13.334°; LT=UT+1) on February 11, 2009 from 9:33 to 9:37 UT. The lower panel shows the SMT results. It can be easily seen that IMFs from 1 to 30 represents the fluctuating part of the signal (δBy), while IMFs from 31 to 82 are the baseline ($By_0$). To distinguish between instrumental origin fluctuations and real signal, a multiscale statistical analysis is needed. For the different scales $\ell$s, we considered the statistics of the values $IMF_\ell(t)$. This technique, called multiscale statistical analysis, calculates and studies the second (the variance σ($\ell$)), the third (the skewness Sk($\ell$)) and the
fourth (the kurtosis excess $K_{ex}(\ell)$=K ($\ell$)−3) moment of the probability distribution p($IMF_\ell$(t)) of $IMF_\ell$(t), and the relative energy $\epsilon_{rel}$, and the Shannon information entropy I($\ell$), respectively defined as:

$$\epsilon_{rel}(\ell) = \frac{\int_\ell |IMF_\ell(t)|^2 dt}{\int_\ell |s(t)|^2 dt}. \qquad \text{Eq. 2}$$

$$I(\ell) = -\sum_{\{IMF_\ell\}} p\big(IMF_\ell(t)\big) \cdot \log_2 p\big(IMF_\ell(t)\big). \qquad \text{Eq. 3}$$

These parameters measure the variability of the statistics of the signal in function of the scale considered [Strumik & Macek,
2008]. Namely: $K_{ex}(\ell)$ indicates how the different $\ell$'s are rich of rare fluctuations [Frisch, 1995]; $\epsilon_{rel}$ measure how "energetically strong" the $\ell$ component is in the Eq. 1. I($\ell$) measures the "degree of randomness" of each $IMF_\ell$(t) component of the signal. In our case the scale $\ell$ corresponds to the peculiar frequency of each IMF of both magnetic and electric field observations.

### 2.4 Instrumental background

We define $IMF_\ell$(t) to have an instrumental origin if two conditions are at the same time satisfied:





1.   The SM test evaluates the $\mathrm{IMF}_\ell(t)$ as a fluctuation;

2.   $K_{ex} (\mathrm{IMF}_\ell(t))$ is almost null and correspondingly $I(\mathrm{IMF}_\ell(t))$ presents a relative maximum.

Indeed, an $\mathrm{IMF}_\ell(t)$ that satisfies these two conditions can be represented as a Gaussian fluctuation characterized by a high "degree of randomness". Thus, it can be identified as instrumental noise. Figure 3 shows an example of a multiscale
statistical analysis of the By component of DEMETER satellite for the same period of Figure 2. Figure 3a shows the $\epsilon_{rel}$ behaviour as a function of the scale $\ell$ (i.e. the frequency). Two energy peaks, at 20 Hz (blue dashed line) and 333 Hz (green dashed line), are clearly visible. Scales lower than 3 Hz have almost null energy (red dashed line), have $K_{ex}(\ell) \sim 0$ and show the highest values of $I(\ell)$. The IMFs corresponding to these scales could be attributed to instrumental noise. Anyway, a more accurate analysis on each IMFs in the interval $\ell < 3$ Hz will be done in the next sections. On the other hand, the IMF related
to 20 Hz are not of instrumental origin because, despite the almost null value of $K_{ex}$, the Shannon entropy shows a concave-upward. In fact, $\ell \cong 20$ Hz is one of the peaks of Shumann Resonance in the extremely low frequency (ELF) portion of the Earth's electromagnetic field spectrum generated and excited by lightning discharges in the cavity formed by the Earth's surface and the ionosphere [Barr et al., 2000 and reference therein]. A similar situation is obtained for $\ell = 333$ Hz. In fact, the relative $K_{ex}(\ell) = 3$ (Figure 3b) and $I(\ell)$ (Figure 3c) show a concave-upward. Thus, the signal associated to 333 Hz is not
originated by instrumental noise.

By the use of those criteria, we can identify all the n<m $\mathrm{IMFs}(\ell)$ originated by instrumental noise. As a consequence, the instrumental background can be defined as:

$$R_b = \sum_{\ell=1}^{n} \mathrm{IMF}(\ell)$$

where $R_b$ is the signal of instrumental origin.

**2.5   Environmental background**

The environmental background has been evaluated through the following steps:

1.   We divided the entire electric and magnetic DEMETER dataset into 2 subsets depending on the local time sector of the satellite orbit (i.e. dayside or nightside). Each subset has been again divided into 2 more subsets characterized by different seismic conditions. The first one ($M_L$) is defined for low seismic activity (M≤3, M being the earthquake
magnitude) and the second ($M_H$) for high seismic activity (M>3). This procedure is crucial to take into account the nature of the earthquake and the different ionospheric response.

2.   Each $M_L$ and $M_H$ subsets were again divided into 3 subsets according to the level of geomagnetic activity. This division is important to take into account possible signals associated with geomagnetic activity. To accomplish this task, we used the Kp index, that is the global geomagnetic storm index and measures the deviation of the most disturbed





horizontal component of the magnetic field at fixed stations worldwide with their own local K index. The K index itself is a three hours long quasi-logarithmic local index of the geomagnetic activity, relative to an unperturbed day curve for the given location. The Kp index ranges from 0 to 9 where a value of 0 means that there is very little geomagnetic activity and a value of 9 means extreme geomagnetic storming. The 3 subsets correspond to three intervals of Kp, that

is: $I_{k,1}$= [0, 2), $I_{k,2}$= [2, 4] and $I_{k,3}$= (4, 9]. $I_{k,1}$, $I_{k,2}$ and $I_{k,3}$ correspond to quiet, moderate and high geomagnetic activity. As a consequence, we finally obtained a total of 12 intervals (hereafter $C_{M,K,L}$, where the subscripts M, K and L correspond to the magnitude interval, Kp interval and local time interval of the satellite orbit, respectively).

3.    The world map has been divided into 1° x 1° latitude/longitude cells. Each $C_{M,K,L}$ will be decomposed by ALIF. For each cell, after the removal of instrumental noise by applying the technique described above, a time – frequency $\epsilon_{rel}$ will

be calculated. Then, a mean $\overline{\epsilon_{rel}}$ will be calculated and stored for each frequency scale. Averaging has been applied only if the ratio $R_\epsilon(\ell) = \frac{\epsilon_{rel}(\ell)}{\overline{\epsilon_{rel}}(\ell)} = 1 \pm 3\sigma(\ell)$, where $\sigma(\ell)$ is the standard deviation of $\overline{\epsilon_{rel}}$ evaluated at the single frequency scale $\ell$.

For each $C_{M,K,L}$, we defined $\overline{\epsilon_{rel}}(\ell)$ as the environmental background. This kind of background gives a representation of both the magnetospheric and ionospheric electric and magnetic fields activity directly driven by the geoelectric and geomagnetic

field variations induced by solar forcing. As a consequence, any distinct signal (over the threshold $1 \pm 3\sigma(\ell)$) could be reasonably studied as an anomalous event.

Figure 4 shows the background components of both the electric (left panels) and the magnetic (right panels) fields over L'Aquila cell ($C_{M,K,L}$) with M<3, Kp<2, L=nightside, which we defined to be the quiet background condition. For the evaluation we used 102 satellite orbits. The results are presented in the satellite reference frame[1] [Berthelier et al., 2006].


### 3. April 4, 2009 case event.

The DEMETER orbit occurred on April 4, 2009 (two days before a 6.3 magnitude earthquake) was identified as anomalous by our technique. In fact, Figure 5, exhibiting $\epsilon_{rel}$ for both the electric and magnetic field components, shows an anomalous signal (s*) at frequency f*=333 Hz, which is not present in quiet conditions (see Figure 4). It is worth noting that

the time of f* onset corresponds exactly to DEMETER passage through L'Aquila geographic footprint. s* has a peculiar e.m. polarization, characterized by a magnetic field oscillating principally in the y-z plane, and an electric field (less clear situation) oscillating principally along the x-y plane (in the satellite reference frame).

Since ALIF extracts both the electric and magnetic field wave forms at each frequency, we were able to calculate the instantaneous phase difference between the two signals, resulting in ~90°. This condition allowed the evaluation of the

Poynting vector $\vec{S} = \vec{E}x\vec{B}$, showing the following characteristic angles with respect to the satellite coordinate system:

---

[1] x is directed along the Nadir direction; z is directed along the satellite velocity vector; y is perpendicular to the x-z plane



$\vartheta_1 = 167.1°$ and $\varphi_1 = 15.4°$ ($\vartheta$ and $\varphi$ being the angles between $\vec{S}$ and x, and $\vec{S}$ and z, respectively). The direction of $\vec{S}$ confirms that s* is directed toward the satellite, coming from ground.

Interestingly, the same peculiar frequency, f*, was found on February 11, 2009, with lower (~60%) $\epsilon_{rel}$ (see Figure 2) and comparable polarization in both magnetic and electric field (not shown). Also for this case event, the evaluated direction of $\vec{S}$ confirms a signal coming from ground ($\vartheta_2 = 154.6°$ and $\varphi_2 = -6.4°$).

## 4. Discussion and conclusions.

The correct identification of a background in the e.m. emission over seismic regions has a crucial role for the detection of possible signals related to earthquake or pre-earthquake activity. The algorithm presented here represents a new and very efficient technique to discriminate among instrumental, environmental and external source signals from satellite observations. The efficiency of ALIF for both non-linear and non-stationary signal analysis, and peculiar frequency onset identification has been proved in several works [i.e., Piersanti and Villante, 2016; Alberti et al., 2016 and reference therein]. Anyway, its possible application to identify correctly the instrumental origin noise has never been presented before. Here, we showed that the coupling between ALIF and MSA represents a powerful tool to identify and remove noise from signal. In fact, our method was able to determine all the noise frequencies declared in electric and magnetic field experiment of DEMETER satellite [Lagoutte et al., 2005], such as the 1 Hz in the E field (see Figure 4). This signal is an effect of the instrumental disturbance, i.e. the sweeping voltage of the Langmuir Probe [Lagoutte et al., 2005]. Concerning the continuous 20 Hz signal detected in the magnetic field observations, we speculated that it can be attributed to one of the peaks of Shumann Resonance in the extremely low frequency (ELF) portion of the Earth's electromagnetic field spectrum generated and excited by lightning discharges in the cavity formed by the Earth's surface and the ionosphere [Barr et al., 2000 and reference therein]. Anyway, Lagoutte et al. [2005] in their DEMETER satellite user guide manual certificated the ~20 Hz as a BANT noise.

On the other hand, this paper presents a useful method for the correct selection of anomalous signals with respect to the evaluated background. The choice of using the ratio $R_\epsilon(\ell)$ was to take into account possible anomalous energy enhancements as well as new signal onset. In addition, as a threshold we used $3\sigma$ in order to make a selection as strong as possible to exclude possible positive false. Anyway, at this stage, a visual inspection of each anomalous signal detected is needed. Last but not least, an analysis of the geomagnetic indices behaviour associated with possible e.m. anomaly detected by our method is crucial. In fact, it is worth remarking that the external origin perturbations, in terms of solar activity, represent the principal disturbance of both the Earth's ionospheric electric and magnetospheric fields [Vellante et al., 2014; Piersanti et al., 2017 and reference therein].

In this context, the 333 Hz component, appearing when the DEMETER flew exactly over L'Aquila (Figure 5), is not visible in the corresponding background (in terms of Kp and M indices - Figure 4) and then may be an interesting anomaly. In fact, the April 4, 2009 was characterized by a very low geomagnetic activity, since the Sym-H index (Sym-H is





the ring current activity index) was between 8 and 10 nT, and AE index (AE is Auroral Electroject activity index) was less than 105 nT. This confirms that the April 4, 2009 was a Solar Quiet (Sq) day [Matsushida and Maeda, 1965; Chulliat et al., 2005]. Sq is caused by the concurring contribution of a current systems flowing in the so called ionospheric dynamo region and of the induced telluric currents in the Earth's upper mantle. Briefly, their interaction gives rise to two pairs of vortices:

two in the sunlit hemisphere and the other two in the dark one [Richmond et al., 1976; Shinbori et al., 2014]. This is confirmed by the behaviour of the geomagnetic field observation at L'Aquila ground station (Figure 6), which presents the typical Sq daily variation at middle/low latitude in April [De Michelis et al., 2010]. In particular, no ELF perturbations are observed between 20:30 UT and 20:40 UT. As a consequence, we can reasonably assert that s* cannot be related to any solar perturbation.

As a matter of fact, the relative Poynting vector $\vec{S}$ indicates a wave propagating from ground to the ionosphere. The s* peculiar polarization might be associated with a horizontal current system flowing at ground, switched on by an anomalous ground impedance generated by the fault-break. It is thought that the low frequency components (ULF/ELF) of seismo electro-magnetic emissions (SEME) waves generated by pre-seismic sources (such as local deformation of field, rock dislocation and micro-fracturing, gas emission, fluid diffusion, charged particle generation and motion, electro-kinetic,

piezo-magnetic and piezoelectric effects, fair weather currents) are transmitted into the near-Earth space [Dobrovolsky, 1989; Teisseyre, 1997; Pulinets and Kirill, 2000; Sorokin, 2001]. During their propagation through the solid crust, the SEME waves characterized by lower period are attenuated. As a consequence, only low frequency waves (in the ULF/ELF band) can go over the Earth's crust and propagate through the ionosphere-magnetosphere system with moderate attenuation [Bortnik and Bleier, 2004]. Observations from Low-Earth-Orbit (LEO) satellite seems to confirm this scenario. In fact, pre-

seismic variations of electric and magnetic fields and of ionospheric plasma temperature and density [Parrot, 1993; Chmyrev, 1997, Buzzi, 2006] have been observed from a few minutes to several hours (2-6 hours) prior to Earthquakes of moderate or strong magnitude (M>4.0). Unfortunately, no magnetotelluric measurements that could confirm or neglect our hypothesis, were available for the event under investigation.

    Interestingly, on February 11, 2009, a similar signal, characterized by lower (~60%) $\epsilon_{rel}$ and comparable

polarization, was observed on both electric and magnetic field components. Despite the direction of $\vec{S}$ confirms that also this signal comes from ground ($\vartheta_2 = 154.6°$ and $\varphi_2 = -6.4°$), nothing can be speculated to its physical causes in this case. In fact, first of all it is characterized by different solar activity conditions with Sym-H between 40 nT and 50 nT, and AE between 150 nT and 200 nT. Last but not least, the satellite orbit was diurnal. Hence, to be consistent with our cell division method, the February 11, 2009 cannot be compared neither to our quiet background nor to the April 4, 2009 case event.

Anyway, its peculiar characteristics needs to be investigated in a companion paper containing a statistical approach.

    The analysis of the April 4, 2009 event, showed that only through a multi-instrumental and multi-disciplinary approach, a reliable disentangle of the earthquake effects from changes due to the physical processes that govern the ionosphere dynamic and to natural EME can be obtained.



This work could be considered as a suggested analysis approach for the forthcoming scientific phase of the first CSES mission (launched on February, 2018, and still in the commissioning phase) aiming to reduce the lack of knowledge in the lithosphere-ionosphere coupling. As soon as further applications, performed on different seismic events, reaches the expected reliability, the proposed method could be used to compute the global background level (with 1 squared degree of resolution) for a direct real-time comparison of CSES in flight data.

**Acknowledgments**

The authors thank F. Capaccioni for useful scientific discussion and comments. The authors want to thank the CNES teams in charge of the DEMETER project and its in-flight operations and D. Lagoutte, J.Y. Brochot and S. Berthelin for their remarkable work at the DEMETER Science Mission Centre. The results presented in this paper rely on the data collected at L'Aquila. We thank INGV (Istituto Nazionale di Geofisica e Vulcanologia), for supporting its operation and INTERMAGNET for promoting high standards of magnetic observatory practice (www.intermagnet.org). Authors also thank the Italian Space Agency (ASI) for the financial support under the contract ASI "LIMADOU scienza" n° 2016-16-H0.

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

20



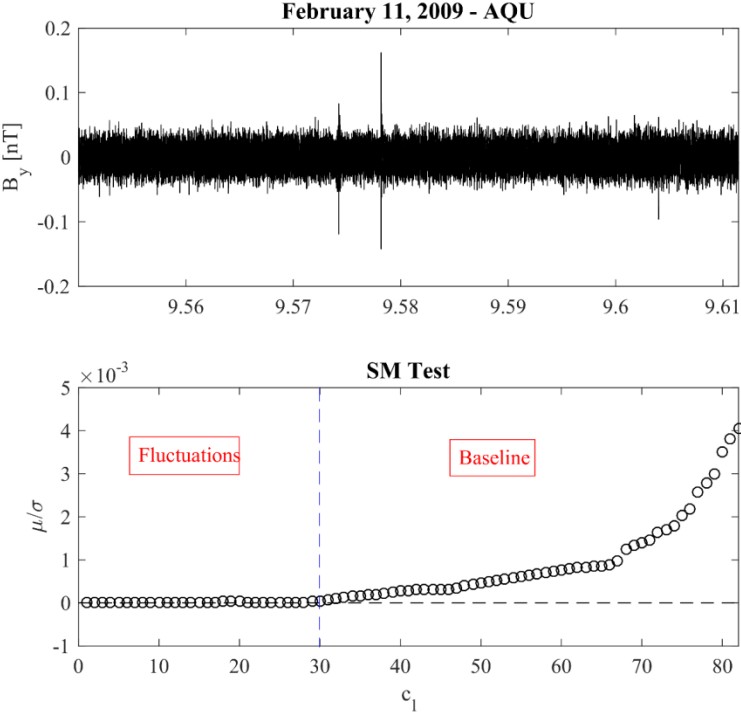

**Figure 2** An example of the application of the SMT test (lower panel) to the magnetic field observations (upper panel) over l'Aquila on February 11, 2009 from 9:33 to 9:39 UT.



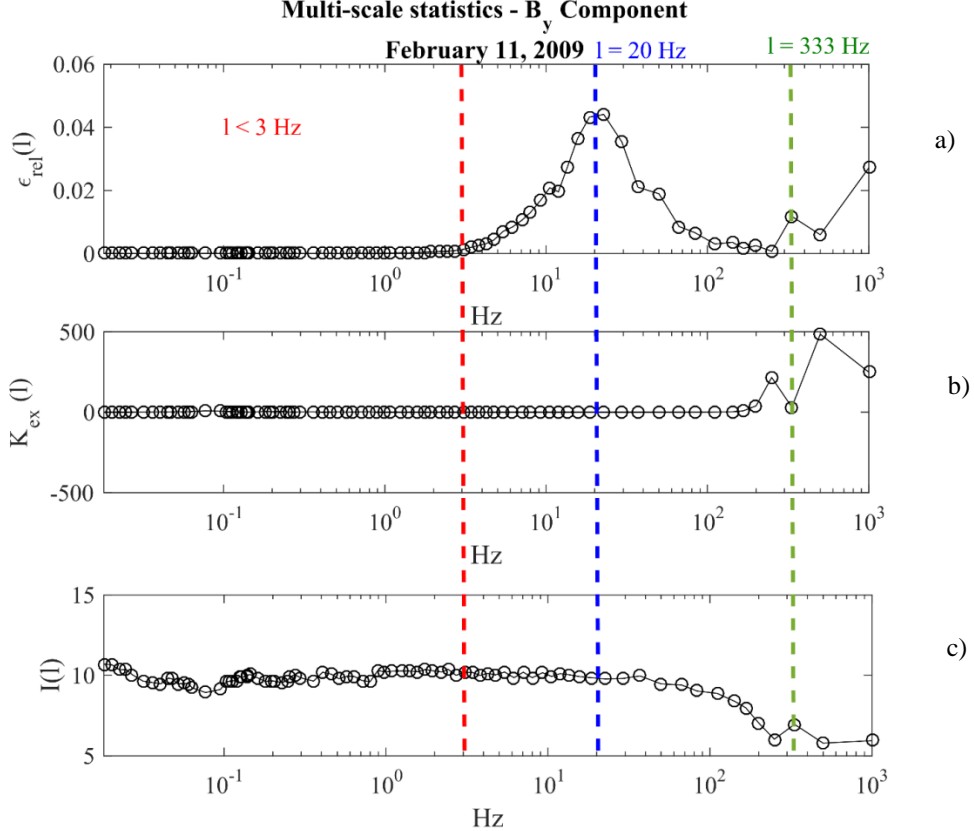

**Figure 3.** Example of a multiscale statistical analysis of the By component for the same data of Figure 2: panel a) $\epsilon_{rel}$ vs. frequency; panel b) $K_{ex}$ vs. frequency; panel c) $I$ vs. frequency. Two energy peaks, at 20 Hz and at 333 Hz are clearly visible.



**Figure 4.** Environmental background for l'Aquila cell as evaluated by ALIF in terms of $\overline{\epsilon_{rel}}(\ell)$ vs. time and frequency for the reference quiet period (M<3, Kp<2, L=nightside). Left panels shows the $\overline{\epsilon_{rel}}(\ell)$ for the three components of the electric field; right panel shows the $\overline{\epsilon_{rel}}(\ell)$ for the three componets of the magnetic field.



**Figure 5:** Anomalous event detected over L'Aquila on April 4, 2009. Left panels shows the $\epsilon_{rel}(\ell)$ vs. time and frequency for the three components of the electric field; right panels shows the $\epsilon_{rel}(\ell)$ vs. time and frequency for the three components of the magnetic field. A clear anomalous energy peak at 333 Hz, with respect to the quiet reference conditions (Figure 4) appears in both magnetic and electric field.





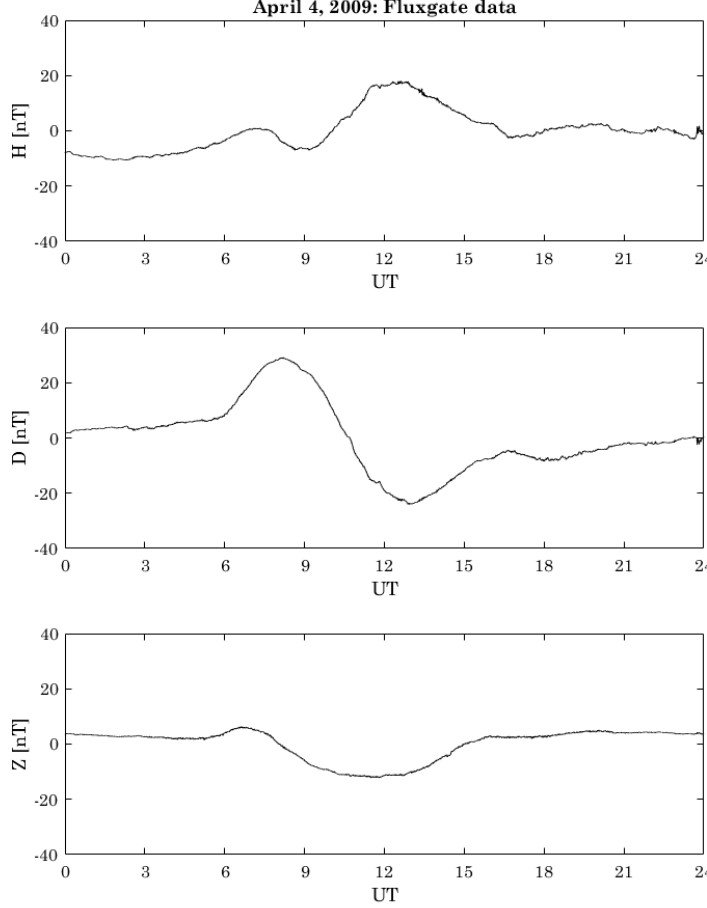

**Figure 6:** Geomagnetic field observations at L'Aquila ground station: upper panel shows the H (North-South) component; middle panel shows the D (East-West) component; lower panel shows the Z (vertical) component. The observations shows the typical Sq daily variations.