# Peer review of "Electromagnetic field observations by the DEMETER satellite in connection with the 2009 L'Aquila earthquake"

_Annales Geophysicae, 2018_

## Referee Comment (RC1) · Anonymous Referee #1 · 15 Jul 2018

This paper is related to the use of an up to date method of wave analysis to reveal anomalies in a data set of electromagnetic measurements. A comparison is done between results with and without events and in case of magnetic activity. The paper is well written and is worthy of publication in AG. I just encourage the authors to apply their method to other EQs in a future publication. Page 1, Line 25 Bell et al. or Bell . In fact there are some problems with the references which are related to the date of the papers (see for example Berthelier et al. noted as 2005 but in fact it is 2006). Page 4 line 9 meaning of SMT ? Page 6 lines 17-18 what means "over l'Aquila" ? Latitude, longitude area ? Page 7 line 22, what is BANT ?

[Figure]

---

## Referee Comment (RC2) · Anonymous Referee #1 · 16 Jul 2018

Concerning the item 4, I want to know the dimension (in latitude and longitude) of the area around L'aquila which has been investigated when you are speaking about 102 half-orbits.

---

## Author Comment (AC1) · 16 Jul 2018

Response to Anonymous Referee #1

We thank the Reviewer who appears to agree with the significance of our results and comments our work as suitable for publication after minor revisions. In the revised version all her/his suggestions have been considered, namely:

1) I just encourage the authors to apply their method to other EQs in a future publication.

Thanks for the suggestion. We are already working on a companion paper about the

application of our technique to other Earthquakes in Italy and in Chile.

2) Page 1, Line 25 Bell et al. or Bell . In fact there are some problems with the references which are related to the date of the papers (see for example Berthelier et al. noted as 2005 but in fact it is 2006). We fixed all the references in the paper.

3) Page 4 line 9 meaning of SMT ? SMT is the acronym of Standardized Mean Test. Anyway, since we referred as SM test in the manuscript, we changed SMT into SM test.

4) Page 6 lines 17-18 what means "over l'Aquila" ? Latitude, longitude area ? We added in the paper the geomagnetic location of the DEMETER satellite flying over l'Aquila.

5) Page 7 line 22, what is BANT ? We added in the revised text the meaning of BANT, that is: Boîtier Analogique et Numérique de Traitement. It is equivalent to the Analog Processing Unit.

---

## Author Comment (AC2) · 16 Jul 2018

We thank the referee for his comments that has been considered, namely:

1) Concerning the item 4, I want to know the dimension (in latitude and longitude) of the area around L'aquila which has been investigated when you are speaking about 102 half-orbits.

We modify the text specifying the dimension of the Area around L'Aquila that we investigated. We considered a square cell centered at the geographic latitude and longitude of L'Aquila of 1° of latitude and 1° of longitude.

---

## Referee Comment (RC3) · Anonymous Referee #2 · 20 Aug 2018

This article shows a particular multiscale and multifrequency analysis (that the Authors call ALIF) of DEMETER electric and magnetic satellite data over the region interested by the 2009 L'Aquila Earthquake, in order to search for specific anomalies that can be related to the impending earthquake. Although the resulting anomaly found around two days before the mainshock seems interesting, no strong evidence is given to support its pre-earthquake origin. However, the ALIF technique is very original and efficient, but I think, it would be applied to many more cases, than just a case, as applied in this work, before to assess its real value in detecting pre-earthquake anomalies.

The present version of the paper is not ready for publication, rather it needs a major

revision in order to clarify all points I indicate below.

Main points

- How unique is the 4 April 2009 anomaly? Were other 3-sigma+ anomalies in the considered epicentral region? What about analyzing a close but different region, in order to make an objective comparison? By the way, an analogous figure of Fig.2 (made for February 11, 2009) but for the 4 April 2009, when the major anomaly has been found, is missing.

- It is not clear the precise size and location of the area of analysis considered in the paper (pag.3, line 14-15), apart from expecting it was surrounding the L'Aquila earthquake epicenter. Neither the interval of time of the data. Apparently, it seems that all orbits from 2004 to 2010 passing over L'Aquila area were considered.

- From the direction of the Pointing vector the Authors affirm that the perturbations producing the found anomalies (on 11 February and 4 April 2009) come from the ground. However it is very strange that the corresponding lithospheric regions at the origin of the two perturbations are quite different (see the different angles). Could the authors explain this difference?

- The only geomagnetic index used in the analysis for discriminating the level of external magnetic activity is Kp. Although this is a good index to understand the overall level of activity, it is only partial. I would suggest to take into account more stringent conditions considering also Dst and AE indices (these indices – Dst in terms of its proxy Sim-H index- are just mentioned at the end of pag.7, the beginning of pag.8, for the case of the 4 April anomaly). In addition, also the behavior of the same indices in the previous 5-6 hours should be considered, because the magnetic activity could be at the stage of recovery phase, after some perturbation affecting initially auroral regions. For instance, Perrone et al. (2018) do not limit their attention only to the 3-hour period of interest, but they also consider AE for all the previous 6 hours in their work (and a daily Ap less than 15), otherwise the possible anomaly is rejected as internal origin.

- No reasonable and clear model of the generation of the 330 Hz frequency at the earthquake preparation is given, and how much it could be related to the L'Aquila mainshock fault and the composing rocks. From the supposed conductivity structure under L'Aquila area, this frequency seems to be largely attenuated by the skin-depth penetration condition avoiding to cross all lithospheric medium from the fault rocks and be transmitted above in the atmosphere.

Minor points

Pag.1, Abstract. Doubts on the use of the term "noise" in this context. Line 13 (also pag.2, line 24) : Cicone et al, 2017 is missing in the references list.

Pag.1 Line 24. "dynamics" better than simply "dynamic".

Pag.1 Line 25. Bell et al., 1982 is not a complete and appropriate reference for the first sentence of the paper. Use other more specific references.

Pag.2 line 4. "between internal and external components"

Pag.3 Line 20. Incongruence between citation Piersanti et al. 2018 and the references indicated as Piersanti line 24)

Pag.3 Line 23. The reference Cohen 2001 is missing.

Pag.3, line 29. SM Test is not defined in this section.

Pag.3 Line 31. Correct "proprieties" in "properties".

Pag.4 Line 6. I do not find any Flandrin et al. 1998. Do you mean just Flandrin 1998?

Pag.4 Line 20. Please correct "measure" in "measures"

Pag.7, Line 22. Please define "BANT noise".

Pag.9 Line 27. Buzzi 2007 is not available at the given link.

Pag.15, Line 7. Please correct "componets" in "components".

Additional Reference

Perrone et al., Ionospheric anomalies detected by ionosonde and possibly related to crustal earthquakes in Greece, Ann. Geophysicae, 36, 361–371, 2018.
* * *

---

## Author Comment (AC3) · 27 Aug 2018

We thank the Reviewer who appears to agree with the significance of our results and comments our work as suitable for publication after major revisions. In the revised version, that we attached here (bold text indicates the text changes), all her/his suggestions have been considered, namely:

Main points:

1. How unique is the 4 April 2009 anomaly? Were other 3-sigma+ anomalies in the considered epicentral region? What about analyzing a close but different region, in or-

der to make an objective comparison? By the way, an analogous figure of Fig.2 (made for February 11, 2009) but for the 4 April 2009, when the major anomaly has been found, is missing. The April 4, 2009 is the only 3-sigma+ anomaly that our method detected during a real solar quiet period. In fact, it was not only characterized by a geomagnetic solar quiet period and a ionospheric quiet condition for the entire day, but it happened exactly when the satellite flew over L'Aquila geographic position. The other anomalies detected by our method were found in correspondence of medium/active solar conditions or in positions that are close to L'Aquila but are not exactly over L'Aquila. Concerning the analysis of a close but different region, we are not sure that it is a correct way to make an objective comparison. In our opinion, Italy is characterized by different soil and magnetotelluric peculiarities that cannot be compared. Anyway, our 1°x1° cell (corresponding to 220km x 160km at L'Aquila geographic position) should be enough large to incorporate different regions. Finally, we added a new figure (new figure 5) in which we put the April 4, 2009 observations (like in figure 2) and the DEMETER orbit.

2. It is not clear the precise size and location of the area of analysis considered in the paper (pag.3, line 14-15), apart from expecting it was surrounding the L'Aquila earthquake epicenter. Neither the interval of time of the data. Apparently, it seems that all orbits from 2004 to 2010 passing over L'Aquila area were considered. We modified the text explaining the size and the location of the analysis considered in the paper and the time interval considered (as shown in the new figure 5). Concerning the environmental and the instrumental background, we considered all the orbits from 2004 to 2010 passing over 1°x1° cell centred in L'Aquila geographic position. However, as expressed in the text, for the April 4, 2009, for the background we considered only the corresponded Solar Quiet background, that has been evaluated with 73 orbit, as expressed in the text.

3. From the direction of the Pointing vector the Authors affirm that the perturbations producing the found anomalies (on 11 February and 4 April 2009) come from the ground.

However it is very strange that the corresponding lithospheric regions at the origin of the two perturbations are quite different (see the different angles). Could the authors explain this difference? We thank the reviewer for the comments, through which we found a typing error in the February 11, 2009 Pointing vector angles. We corrected the $\varphi$ direction. In the first version of the paper we wrote a negative number, while it was positive. As you can see, the S values are comparable and probably they came from the same lithospheric region. In this case, the possible explanation of the little difference in the two S values could be related to the different orbit of DEMETER satellite passing through our cell: the April one is exactly over L'Aquila (see the new Figure 5), the other one is more eastward with respect to L'Aquila position. Anyway, as expressed in the paper, we stressed that these events cannot be comparable because, as stated in the paper, the February event was characterized by different solar conditions that prevent any possible lithospheric explanation (as you also stated in the next objection).

4. The only geomagnetic index used in the analysis for discriminating the level of external magnetic activity is Kp. Although this is a good index to understand the overall level of activity, it is only partial. I would suggest to take into account more stringent conditions considering also Dst and AE indices (these indices – Dst in terms of its proxy Sim-H index- are just mentioned at the end of pag.7, the beginning of pag.8, for the case of the 4 April anomaly). In addition, also the behavior of the same indices in the previous 5-6 hours should be considered, because the magnetic activity could be at the stage of recovery phase, after some perturbation affecting initially auroral regions. For instance, Perrone et al. (2018) do not limit their attention only to the 3-hour period of interest, but they also consider AE for all the previous 6 hours in their work (and a daily Ap less than 15), otherwise the possible anomaly is rejected as internal origin. We again thanks the reviewer for the interesting suggestion. We calculated again both the environmental and instrumental background using both Sym-H and AE indices for the evaluation of the geomagnetic conditions. We added a full explanation in the text. As you can see, we used very restrictive conditions for the SQ identification, corresponding to Sym-H= [10 nT, -10 nT) and AE<100 nT. Anyway, our results did not changed and the

April 4, 2009 anomaly was found again. We added the Perrone et al.[2018] reference in the paper.

5. No reasonable and clear model of the generation of the 330 Hz frequency at the earthquake preparation is given, and how much it could be related to the L'Aquila main-shock fault and the composing rocks. From the supposed conductivity structure under L'Aquila area, this frequency seems to be largely attenuated by the skin-depth penetration condition avoiding to cross all lithospheric medium from the fault rocks and be transmitted above in the atmosphere. In general we agree with the Reviewer. In fact, we proposed a possible explanation of the 333 Hz EME detected by our algorithm, if it was related to the earthquake preparation. But we are not sure about it, and we stressed it in the paper. The problem is that both our explanation and your "counter-hypothesis" depend on the conductivity structure. The attenuation model of the frequency under a focal area strictly depends on the conductivity values, that needs to be measured through magnetotelluric observations. In fact, this is final remark of our paper (pag. 8, lines 26-28 and pag. 9, lines 3-5). We are sure that only through a multi-instrumental and multi-disciplinary approach, a reliable disentangle of the earthquake effects from changes due to the physical processes that govern the ionosphere dynamic and to natural EME can be obtained.

Minor points:

Pag.1, Abstract. Doubts on the use of the term "noise" in this context. Line 13 (also pag.2, line 24) : Cicone et al, 2017 is missing in the references list: We changed the term "noise" into background and we added Cicone et al., 2017 in the reference list.

Pag.1 Line 24. "dynamics" better than simply "dynamic": Change made.

Pag.1 Line 25. Bell et al., 1982 is not a complete and appropriate reference for the first sentence of the paper. Use other more specific references: We added more references.

Pag.2 line 4. "between internal and external components": Change made.

Pag.3 Line 20. Incongruence between citation Piersanti et al. 2018 and the references indicated as Piersanti line 24): We fixed the incongrunce in the reference list.

Pag.3 Line 23. The reference Cohen 2001 is missing: reference added.

Pag.3, line 29. SM Test is not defined in this section: we modified the section title in order to define the acronym SM.

Pag.3 Line 31. Correct "proprieties" in "properties": Change made.

Pag.4 Line 6. I do not find any Flandrin et al. 1998. Do you mean just Flandrin 1998?: Change made.

Pag.4 Line 20. Please correct "measure" in "measures": Change made.

Pag.7, Line 22. Please define "BANT noise": We added in the revised text the meaning of BANT, that is: Boîtier Analogique et Numérique de Traitement. It is equivalent to the Analog Processing Unit.

Pag.9 Line 27. Buzzi 2007 is not available at the given link: We added Buzzi 2007 in the reference list.

Pag.15, Line 7. Please correct "componets" in "components": Change made.

Additional Reference: Perrone et al., Ionospheric anomalies detected by ionosonde and possibly related to crustal earthquakes in Greece, Ann. Geophysicae, 36, 361–371, 2018: We added the reference in the paper.

Please also note the supplement to this comment:
https://www.ann-geophys-discuss.net/angeo-2018-67/angeo-2018-67-AC3-supplement.pdf

**Supplement:**

[revised manuscript text omitted]

---

## Referee Report (RR1)

**Comments to revised version of Bertello et al. "Electromagnetic field observations by the DEMETER satellite in connection with the 2009 L'Aquila earthquake"**

The present version has been improved. However I am still unconvinced about the possibility of the 4 April 2009 anomaly to be a precursor of the earthquake then occurred on 6 April 2009.

**The present version of the paper is not ready for publication, rather it needs a more detailed analysis of confutation.**

**Main points**

**( I show again my comments already presented in my previous review in italics. This because they were not actually considered fully).**

- *"What about analyzing a close but different region, in order to make an objective comparison? "* My previous request was/is fundamental because we need a confutation case to establish that what is seen on 4 April was only occurring above L'Aquila and not elsewhere.
- *"In addition, also the behavior of the same indices in the previous 5-6 hours should be considered"* It is true that at the time of the found anomaly AE was little, but 5-6 hours before it was not (http://wdc.kugi.kyoto-u.ac.jp/ae_provisional/200904/index_20090404.html). So it would be more reasonable to attribute the anomaly to the electric field effects from auroral latitudes to middle latitudes . To possibly exclude this case, it should be shown that the track before and after L'Aquila region did not present the same anomaly.

---

## Author Response (AR2)

**Response to Reviewer #1**

We thank the Reviewer who appears to agree with the significance of our results and comments our work as suitable for publication as it is.

**Response to Reviewer #2**

We thank the Reviewer who appears to agree with the significance of our results and comments our work as suitable for publication after a more detailed analysis of confutation. In the revised version all her/his suggestions have been considered, namely:

**Main points**

- **"What about analysing a close but different region, in order to make an objective comparison? "**
  **My previous request was/is fundamental because we need a confutation case to establish that what is seen on 4 April was only occurring above L'Aquila and not elsewhere.**

  We added here (and not in the paper, where we added few sentences about it) the figures of a close but different region analysis for both the electric (right panels) and magnetic field (left panels) components.

[Figure]

**Figure 1.** Left panels shows the ε_rel (l) vs. time and frequency for the three components of the magnetic field; right panels shows the ε_rel (l) vs. time and frequency for the three components of the electric field.

The latitudinal range of the region is 45.7°< λ < 48.3°, that corresponds to the closest L'Aquila Cell analysed in the manuscript. As the reviewer can see, there is no anomaly centred at frequency of 333Hz as found above L'Aquila. In addition, the analysis of the cell centred at lower latitudes than L'Aquila (not shown) confirmed the same behaviour. So we are confident that what is seen on 4 April was only occurring above L'Aquila and not elsewhere.

- • **- "In addition, also the behaviour of the same indices in the previous 5-6 hours should be considered" It is true that at the time of the found anomaly AE was little, but 5-6 hours before it was not (http://wdc.kugi.kyoto-u.ac.jp/ae_provisional/200904/index_20090404.html). So it would be more reasonable to attribute the anomaly to the electric field effects from auroral latitudes to middle latitudes . To possibly exclude this case, it should be shown that the track before and after L'Aquila region did not present the same anomaly.**

[Figure]

**Figure 2.** Solar Wind parameters and high latitudes indices (left panels); the map of Auroral Oval (AO) emission (in kilo-Röntgen) taken at UT 13:49 and at UT 20:34

To answer the reviewer comment, we added a figure here where we plot (on the left) the principal Solar Wind (SW) parameters together with the AE, AL and AU indices (as downloaded from NASA cdaweb site), and the map of Auroral Oval (AO) emission (in kilo-Röntgen) taken at UT 13:49 (the time of maximum excursion of AE index) and at UT 20:34 (the time of the DEMETER orbit analysed over l'Aquila). The SW observations shows that the sudden peak in the AE index is essentially due the flip of the Bz Interplanetary magnetic field component from positive to negative coupled with an greater than average SW density (10-12 amu/cm$^3$), probably related to a small flux-rope coming from the Sun. This particular structure coming from the Sun gave rise to a particle precipitation into the Ionosphere, as confirmed by the AL index peak whose behaviour follows closely the AE index. Anyway, the length of the SW perturbation is too small to produce a substorm affecting the low-middle latitudes (such as L'Aquila). Indeed, left panels, show that the borders of the AO (red dashed liens) is confined at high latitudes (between 65° and 75° - geomagnetic latitudes) either during the AE peak and during the DEMETER observations. Interestingly, during the AE peak, the southern hemisphere shows a clear substrom activity in the night sector as expected. While during the period analysed in the manuscript, there is not any particular auroral activity, as confirmed by the low AE index during those hours (http://wdc.kugi.kyoto-u.ac.jp/ae_provisional/200904/index_20090404.html). In addition, the L'Aquila geomagnetic trace (figure 7 of the manuscript) did not show any characteristics of substorm activity at that latitude for the entire day, such as Pi2 wave activity or geomagnetic bay ([Olson, J. V. (1999), Pi2 pulsations and substorm onsets: A review, J. Geophys. Res., 104(A8), 17499–17520, doi: 10.1029/1999JA900086.; Sastri, J H, Effect of magnetic storms and substorms on the low- latitude/ equatorial ionosphere, ILWS Workshop on Solar Influence on the Heliosphere and Earth's Environment, Goa, India, 19 - 24 Feb 2006, pp.361-368]). At our knowledge, to see the effects of substorms at L'Aquila latitudes, the AE index should overcome values of 600 nT [Sastri, J H, Effect of magnetic storms and substorms on the low- latitude/ equatorial ionosphere, ILWS Workshop on Solar Influence on the Heliosphere and Earth's Environment, Goa, India, 19 - 24 Feb 2006, pp.361-368; Piersanti et al, 2017, Comprehensive Analysis of the Geoeffective Solar Event of 21 June 2015: Effects on the Magnetosphere, Plasmasphere, and Ionosphere Systems, Solar Physics, 292:169 DOI 10.1007/s11207-017-1186-0]. Finally, figure 1 here shows that both the electric and magnetic field presented almost the same behaviour of the environmental background evaluated over L'Aquila cell (figure 4 of the manuscript), as expected for its close position.

A sentence about the presented discussion has been added to the manuscript in order to avoid the attribution of the anomaly to the electric field effects from auroral latitudes to middle latitudes.

[revised manuscript text omitted]